# Differential Pneumococcal Growth Features in Severe Invasive Disease Manifestations

Daan W. Arends,[a] Wynand Alkema,[b,c] Indri Hapsari Putri,[a,d] Christa E. van der Gaast–de Jongh,[a] Marc Eleveld,[a] Jeroen D. Langereis,[a] Quirijn de Mast,[d] Jacques F. Meis,[e] Marien I. de Jonge,[a] Amelieke J. H. Cremers[a,f*]

[a]Laboratory of Medical Immunology, Radboud Institute of Molecular Life Sciences, Radboudumc Center for Infectious Diseases, Radboudumc, Nijmegen, The Netherlands
[b]TenWise BV, Leiden, The Netherlands
[c]KCBBE, Institute for Life Science & Technology, Hanze University of Applied Sciences, Groningen, The Netherlands
[d]Department of Internal Medicine, Radboud Institute of Health Sciences, Radboudumc Center for Infectious Diseases, Radboudumc, Nijmegen, The Netherlands
[e]Department of Medical Microbiology and Infectious Diseases, Canisius-Wilhelmina Hospital, Nijmegen, The Netherlands
[f]Department of Medical Microbiology, Radboudumc Center for Infectious Diseases, Radboudumc, Nijmegen, The Netherlands

Marien I. de Jonge and Amelieke J. H. Cremers contributed equally to this article.

**ABSTRACT** The nasopharyngeal commensal *Streptococcus pneumoniae* can become invasive and cause metastatic infection. This requires the pneumococcus to have the ability to adapt, grow, and reside in diverse host environments. Therefore, we studied whether the likelihood of severe disease manifestations was related to pneumococcal growth kinetics. For 383 *S. pneumoniae* blood isolates and 25 experimental mutants, we observed highly reproducible growth curves in nutrient-rich medium. The derived growth features were lag time, maximum growth rate, maximum density, and stationary-phase time before lysis. First, the pathogenicity of each growth feature was probed by comparing isolates from patients with and without marked preexisting comorbidity. Then, growth features were related to the propensity of causing severe manifestations of invasive pneumococcal disease (IPD). A high maximum bacterial density was the most pronounced pathogenic growth feature, which was also an independent predictor of 30-day mortality ($P = 0.03$). Serotypes with an epidemiologically higher propensity for causing meningitis displayed a relatively high maximum density ($P < 0.005$) and a short stationary phase ($P < 0.005$). Correspondingly, isolates from patients diagnosed with meningitis showed an especially high maximum density and short stationary phase compared to isolates from the same serotype that had caused uncomplicated bacteremic pneumonia. In contrast, empyema-associated strains were characterized by a relatively long lag phase ($P < 0.0005$), and slower growth ($P < 0.005$). The course and dissemination of IPD may partly be attributable to the pneumococcal growth features involved. If confirmed, we should tailor the prevention and treatment strategies for the different infection sites that can complicate IPD.

**IMPORTANCE** *Streptococcus pneumoniae* is a leading infectious cause of deaths worldwide. To understand the course and outcome of pneumococcal infection, most research has focused on the host and its response to contain bacterial growth. However, bacterial epidemiology suggest that certain pneumococcal serotypes are particularly prone to causing complicated infections. Therefore, we took the bacterial point of view, simply examining *in vitro* growth features for hundreds of pneumococcal blood isolates. Their growth curves were very reproducible. Certain poles of pneumococcal growth features were indeed associated with specific clinical manifestations like meningitis or pleural empyema. This indicates that bacterial growth style potentially affects the progression of infection. Further research on bacterial growth and adaptation to different host environments may therefore provide key insight into pathogenesis of complicated invasive disease. Such knowledge could lead to more tailored vaccine targets or therapeutic approaches to reduce the million deaths that are caused by pneumococcal disease every year.

Address correspondence to Amelieke J. H. Cremers, melieke.cremers@radboudumc.nl.

*Present address: Amelieke J. H. Cremers, Department of Fundamental Microbiology, University of Lausanne, Lausanne, Switzerland.

The authors declare no conflict of interest.

**KEYWORDS** bacterial growth, *S. pneumoniae*, serotype, empyema, meningitis, pathogenesis

*S*treptococcus pneumoniae asymptomatically colonizes the upper respiratory tract, from which it is transmitted to other hosts. It can also cause life-threatening infections such as pneumonia and meningitis (1). In 2015, the incidence of invasive pneumococcal disease (IPD) in Europe and the United States was 6 and 9 per 100,000, respectively (2, 3). However, the highest morbidity and mortality from pneumococcal infections occurs in low- and middle-income countries (4). Pneumococcal conjugate vaccines are effective at preventing IPD in both children and adults (5–7), but the incidence of IPD in the elderly has not decreased due to replacement by nonvaccine serotypes (8), which is why *S. pneumoniae* remains a primary cause of bacterial pneumonia and meningitis globally (9, 10). Mortality from pneumococcal meningitis ranges from 16 to 37% in developed countries and up to 51% in resource-poor settings (11). Furthermore, half of the surviving patients have neurological sequelae such as focal neurological deficits and cognitive impairment (12). Pleural empyema is another important complication of pneumococcal pneumonia that may require surgical drainage and is associated with a heavily protracted course of disease and recovery (13–15).

The pathogenesis of pneumococcal meningitis or empyema involves both host and bacterial characteristics (16, 17). The pneumococcal capsule is essential for bacterial survival in invasive disease (18, 19). As current pneumococcal vaccines provide serotype-based protection, population dynamics of the >100 different circulating capsule types (serotypes) are globally under surveillance. In epidemiological studies, differences in the propensity to cause severe invasive disease manifestations have been noted. Serotypes 6A, 6B, 6C, 7F, 10A, 15B, 19F, 23A, 23B, and 23F were relatively frequently cultured from cerebrospinal fluid, compared to only blood, while the opposite was observed for serotypes 4 and 14 (20–23). Serotypes relatively frequently involved in pleural empyema were 1, 3, 7F, 8, 9V, 14, and 19A (22, 24–26). Correspondingly, after the introduction of the PCV7 (the 7-valent Pneumococcal conjugate vaccine), several of these serotypes expanded along with a temporary increase in empyema cases observed, up to a vaccine expansion to PCV13, which again repressed these empyema-associated serotypes (27, 28). Previous *in vitro* studies suggest that the presence and the composition of a polysaccharide capsule directly affects pneumococcal growth (29–31). Furthermore, within the population, pneumococcal serotypes often represent particular genetic lineages (32), which may explain why particular growth features are shared among their members.

Many studies have addressed host variants mediating susceptibility and outcome (33, 34), yet there are limited data on the role of bacterial factors. While pneumonia is the major contributor to invasive pneumococcal disease, about one in ten IPD cases involves meningitis. Pneumococcal meningitis primarily originates from *S. pneumoniae* crossing the blood-brain barrier after hematogenous spread, but pneumococci can also invade directly from the upper respiratory tract if there is a breach toward the subarachnoidal space (35). Either way, the potential to cause meningitis requires the pneumococcus to cross different host tissues and to adapt to conditions present in cerebrospinal fluid. The same holds true for *S. pneumoniae* migrating from lung alveoli, through the pleural mesothelium into the pleural cavity where the inflammatory response leads to neutrophil influx and fibrosis (17). Growth of bacterial populations is dynamic and follows environmental cues (36). Moreover, individual bacterial strains display intrinsic growth features in response to changing circumstances, as recently also demonstrated for *S. pneumoniae* exposed to different environmental conditions such as temperature and oxygen (37). In relation to human infection, a short serotype-specific lag phase in nutrient-poor conditions *in vitro* (mimicking the nasopharynx) correlated with an epidemiologically high carriage prevalence (13). However, whether pneumococcal growth features may play a role in the progression from infection to complicated disease and poor outcome has not been studied before.

We propose that the characteristics of three growth phases of *S. pneumoniae* may mediate the development of severe invasive pneumococcal disease manifestations. In

nutrient-rich conditions, like the bloodstream, the speed of exponential growth and the maximum density of *S. pneumoniae* will determine the degree at which organs are exposed to bacteria. For example, at the blood-brain barrier, stochastically, high numbers of bacteria would increase the chance of crossing the barrier, leading to the onset of meningitis (38). Initial entry into the subarachnoid space will not pose an immediate threat to pneumococcal growth, because immune cells still need to be recruited. It likely requires adjustment to nutrient availability before growth can be continued. We hypothesize that a short preparatory lag phase represents high adaptability of the pneumococcus and greater potential to establish meningitis or empyema. Ongoing bacterial infection inside a semiclosed compartment like the meninges or pleural cavity will lead to depletion of favorable growth conditions, followed by a stringent response of the pneumococcal population and a stationary growth phase. Here, prolonged maintenance of intact pneumococcal cells (i.e., postponement of the final lytic phase) may prevent the release of bacterial trigger molecules and delay the initiation of an inflammatory host response leading to clearance (39, 40). In summary, we reason that severe disease would be more likely to establish from pneumococcal strains that show relatively rapid growth, a short lag, and a long stationary phase. Here, we compared pneumococcal growth curves of >400 invasive disease isolates and artificial capsule variants, to investigate whether particular growth features were related to the development of severe disease manifestations.

## RESULTS

**Intrinsic growth features.** We measured *in vitro* growth kinetics for 383 consecutive *S. pneumoniae* blood isolates collected from mainly adult IPD patients and for 25 experimental mutants (artificial capsule switch and capsule deletion). Growth curves for individual pneumococcal strains were highly reproducible (Fig. 1). For each growth feature, measurement errors were fractional in comparison to the diversity across the cohort studied (Fig. S1). No strong relatedness between the four growth features could be identified, with a maximum $r^2$ of 0.04 for the pair growth speed and maximum density (Fig. S2 and S3). Together, these results indicated that pneumococcal strains held intrinsic growth patterns and that growth features could vary independently from the other phases of the growth curve.

**Direct clinical associations.** Next, we examined in what way the determined isolate growth features were related to the observed clinical phenotypes. "Pathogenic poles" of growth features were identified by probing which end of the distribution affected IPD patients with relatively good preexistent health, compared to those who were likely more susceptible due to comorbid and immunocompromising conditions. We considered the growth features associated with healthy hosts as more pathogenic. We observed that infection of relatively healthy adults did not occur by pneumococci that deviated from a short lag phase, high growth rate and maximum density, and a short stationary phase (Fig. 2). Of the isolates studied, 371 (97%) were collected from adult cases, 30 (8%) were involved in pleural empyema, 38 (10%) were involved in meningitis, and 43 (12%) of patients did not survive this episode of IPD. None of the patients suffered from pleural empyema in combination with meningitis. Patients who did not survive their episode of IPD had been infected with isolates that reached higher maximum densities *in vitro* (mean $\pm$ SD $OD_{620}$ = 0.751 $\pm$ 0.005 versus 0.735 $\pm$ 0.003; $P$ = 0.002) (Fig. 3). High maximum pneumococcal density contributed to lethality independently from age and type of infection (Table 1). No other direct association between growth feature and complicated disease (meningitis, empyema, or death) was observed for the IPD population as a whole.

**Serotype-based effects.** Thirty-three different serotypes were represented in the clinical cohort. For one isolate, the serotype could not be determined by Quellung reaction or PCR. For each growth feature, significant variation was observed across serotypes (one-way analysis of variance [ANOVA], $P < 0.0001$ for lag phase, growth rate, and maximum density; Kruskal-Wallis, $P < 0.0001$ for stationary phase) (Fig. S4). Some notable serotype-specific growth characteristics were the universally low maximum growth speed for serotype 3 and

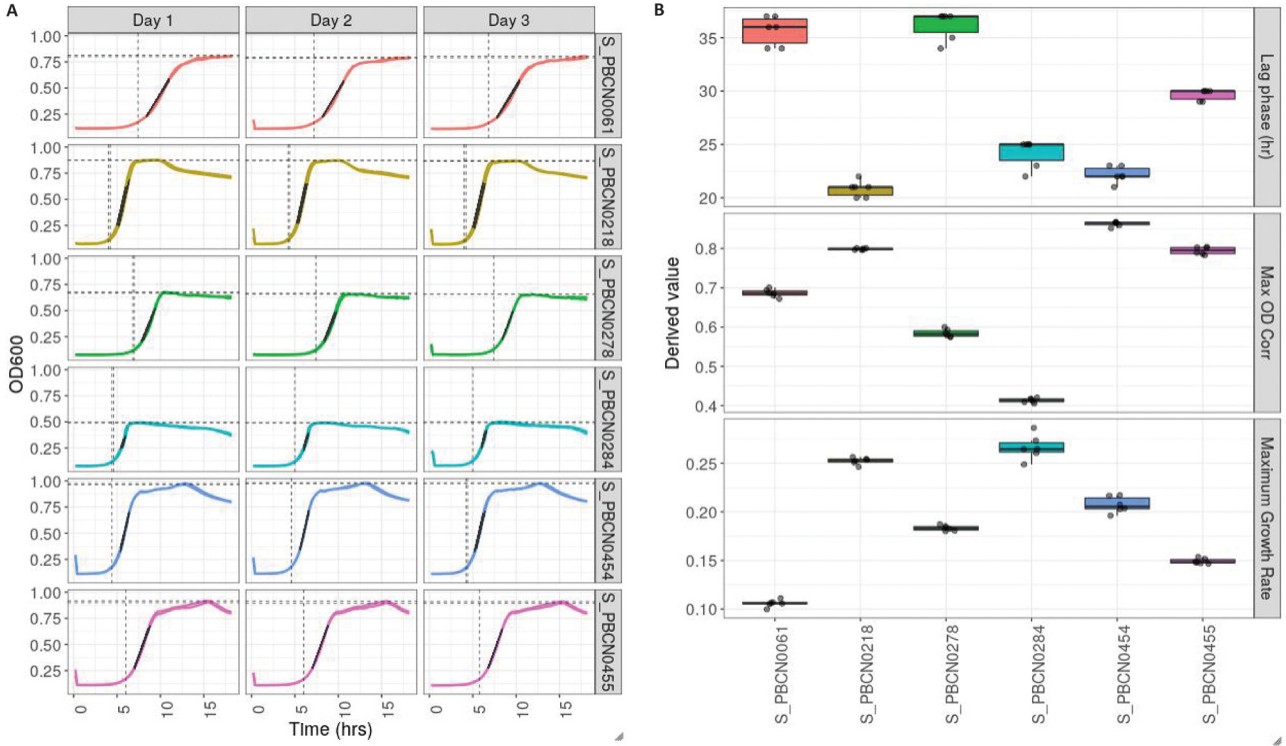

**FIG 1** Intrinsic growth features. For six blood culture isolates of *S. pneumoniae* (in different colors), six repeat measurements are displayed as growth curves with duplicates on three separate days (A) and their derived growth features as box plots (B). Visualizations are downloaded from the online interactive Growth Viewer database (https://fairdb.tenwiseservice.nl/GrowthViewer/). OD₆₀₀, optical density at 600 nm.

the short stationary phase for serotype 8. Experimental deletion of the capsule operon in five pneumococcal isolates did not systematically affect growth features (Fig. S5). Artificial capsular switch mutants from wild-type serotype 4 toward 19 different capsular serotypes demonstrated a universal reduction in growth speed and stationary phase (Fig. S6B and D), the two growth phases that are strongly affected by polysaccharide biosynthesis. The strikingly slow growth speed of serotype 3 isolates in the IPD population was not observed when the serotype 3 capsule was expressed on a serotype 4 background (laboratory strain named TIGR4). The lag phase and maximum density were not heavily or unidirectionally affected by capsular switches (Fig. S6A and C), suggesting that the capacity of TIGR4 to adapt and exploit the rich blood-like substrate may vary by capsular serotype expressed.

**Meningitis.** Serotypes with an epidemiological propensity for causing meningitis (serotypes 6A, 6B, 6C, 7F, 10A, 15B, 19F, 23A, 23B, and 23F) showed a relatively high maximum density (Fig. 4C), and a significant linear trend across the three groups was observed (ANOVA linear trend $P = 0.0006$). In contrast to what we postulated, a long stationary phase was observed for serotypes with a low propensity of causing meningitis (Fig. 4D).

Next, we compared growth characteristics of pneumococcal blood isolates from patients who actually developed meningitis with isolates of the same serotype from patients with a uncomplicated pneumonia. Almost all of the major meningitis-causing serotypes in our IPD cohort followed an epidemiological trend. Within serotypes 6A, 6B, 8, 18C, and 23F, isolates from meningitis cases showed a relatively short lag phase and/or relatively high maximum density compared to sole pneumonia cases (Fig. 5A and B). A markedly short stationary phase was observed for isolates from meningitis cases within the remaining major meningitis-causing serotypes 1, 3, and 23A (Fig. 5C). The only major meningitis-causing serotype in our IPD cohort that opposed an epidemiological trend was serotype 7F, with long lag phases for meningitis isolates (Fig. 5A).

**Empyema.** Serotypes with an epidemiological propensity for causing empyema (1, 3, 7F, 8, 9V, 14, and 19A), showed a pronounced long lag phase (Fig. 6A). The on-average lower growth speed and shorter stationary phase in empyema-associated serotypes were

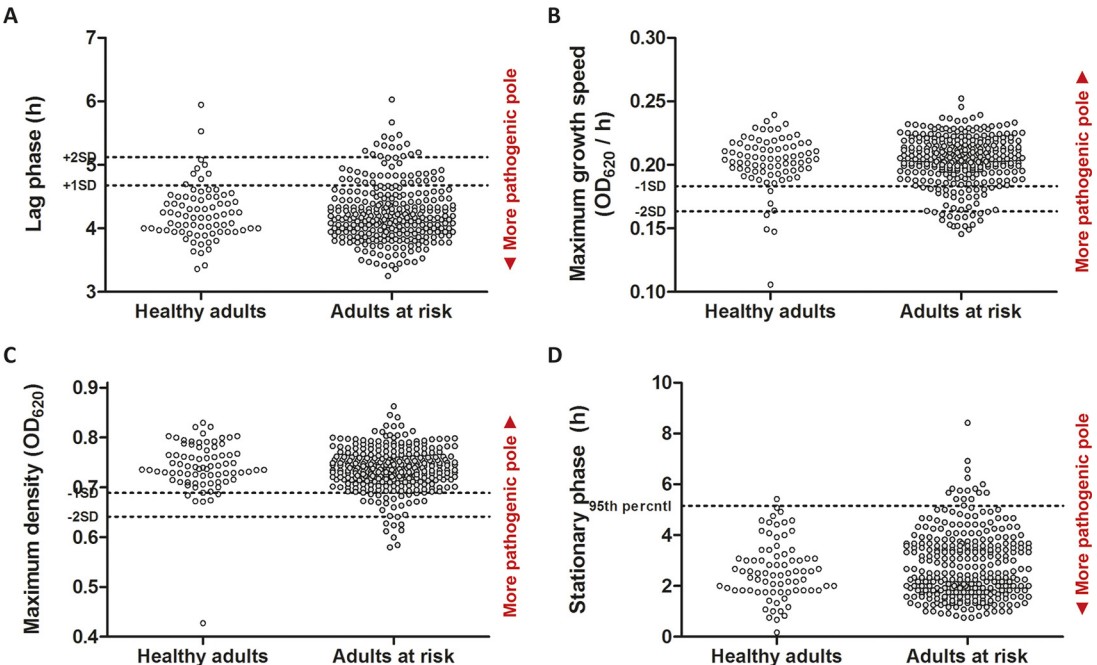

**FIG 2** Pathogenic poles of growth features. Derived growth features for 367 *S. pneumoniae* blood culture isolates that had infected previously relatively healthy adults (*n* = 78) compared to relatively frail adults (*n* = 289) (Charlson comorbidity index score ≤ 2 versus >2). Each dot represents six repeat measurements for one isolate from a corresponding patient. The displayed growth features are lag phase (A), growth speed (B), maximum density (C), and stationary phase (D). $OD_{620}$, optical density at 620 nm.

largely attributable to eccentric properties of serotype 3 and serotype 8, respectively (Fig. 6B and D) but remained a significant feature among the other empyema-causing serotypes.

Within all major empyema-causing serotypes, lag phases were equally long in isolates from empyema cases as from uncomplicated pneumonia cases (Fig. S7A). However, within major empyema-causing serotypes, isolates from empyema cases were differentiated from pneumonia cases by relatively long stationary phases (serotypes 7F, 14, 19A, and 23F) (Fig. S7B).

A tabular overview of all results in relation to the "pathogenic poles" of the growth features is provided in Table 2. Compared to vaccine serotypes, isolates of serotypes that are not targeted by the 13-valent pneumococcal conjugate vaccine demonstrated a lower maximum density but also a shorter stationary phase (Fig. S8).

## DISCUSSION

Our present findings show that invasive pneumococcal disease (IPD) isolates displayed fixed growth curves in rich medium, with marked variation among them. Mortality from IPD was

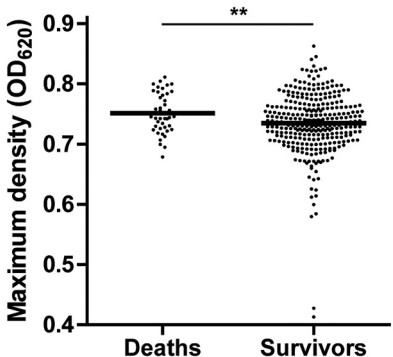

**FIG 3** Maximum density and invasive pneumococcal disease (IPD) mortality. Maximum density reached *in vitro* by *S. pneumoniae* blood culture isolates from patients who died and survived the episode of invasive pneumococcal disease. Each dot represents six repeat measurements for one isolate from a corresponding patient. $OD_{620}$, optical density at 620 nm; **, $P < 0.005$.

**TABLE 1** Independent determinants of death from invasive pneumococcal disease (IPD), including *in vitro* pneumococcal growth feature[a]

| Determinant | Adjusted odds ratio | 95% CI | P value |
|---|---|---|---|
| Age (yr) | 1.06 | 1.02 to 1.08 | 0.0001 |
| Meningitis | 4.84 | 1.80 to 13.04 | 0.002 |
| Maximum pneumococcal density (0.01 $OD_{620}$) | 1.10 | 1.01 to 1.19 | 0.029 |

[a]CI, confidence interval; $OD_{620}$, optical density at 620 nm.

directly associated with isolates that reached a high growth density *in vitro*. While vigorous pneumococcal growth was associated with meningitis, empyema may require indolent growth.

This is the first study to systematically examine pneumococcal growth features for human blood culture isolates. The size of the study cohort made it representative for the spectrum of serotypes and clinical syndromes involved. Additional artificial pneumococcal mutants enabled us to investigate the contribution of the serotype to growth features. The categorization of serotypes and clinical phenotypes holds a certain degree of uncertainty. As an example, in several studies (like in our study region), serotype 1 bacteremia was infrequently associated with meningitis, while in the African meningitis belt, serotype 1 is still responsible for the majority of pneumococcal meningitis cases (41). The clinical phenotypes were determined by the attending physician, and residual complicated IPD can never be fully excluded. Also, we need to consider the possibility that more vigorous bacterial growth features increase the likelihood of recognizing severe infections as they may become more clinically manifest via a stronger host response, but also in terms of detection by focal cultures.

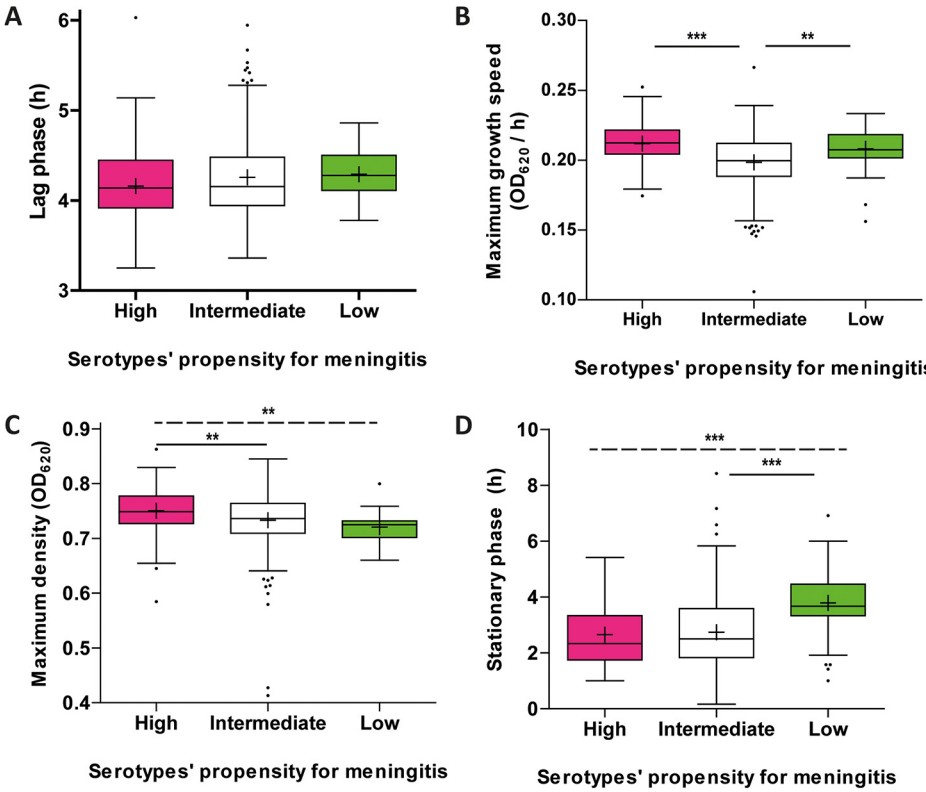

**FIG 4** Serotypes' epidemiological propensity for causing meningitis. Derived growth features for 383 *S. pneumoniae* blood culture isolates stratified according to the serotypes' epidemiological propensity for causing meningitis (high, *n* = 108, pink; intermediate, *n* = 234, white; low, *n* = 41, green). Distributions are visualized as either Tukey box plots with mean (+) and outliers (·). The displayed growth features are lag phase (A), growth speed (B), maximum density (C), and stationary phase (D). The dashed horizontal line indicates a trend across the three categories. $OD_{620}$, optical density at 620 nm; *, *P* < 0.017 (Bonferroni corrected significance threshold); **, *P* < 0.005; ***, *P* < 0.0005.

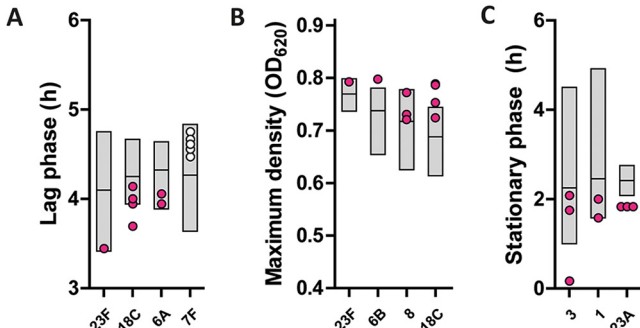

**FIG 5** Within-serotype differentials of meningitis cases. Derived growth features for *S. pneumoniae* blood culture isolates. The data are displayed for meningitis cases that show marked polarity for a growth feature within their serotype. Gray boxes represent isolates from patients with uncomplicated pneumonia caused by that serotype (from minimum to maximum, with horizontal bar at mean respiration median value), while pink dots represent isolates from patients with confirmed meningitis. The displayed growth features are lag phase (A), maximum density (B), and stationary phase (C). $OD_{620}$, optical density at 620 nm.

As we were interested to determine differences in intrinsic growth features in an initial blood-like environment, growth curves were measured in rich culture medium. We did not use a chemostat, but instead allowed for nutrient depletion over time, because this will also happen in a more compartmental infection of the subarachnoid or pleural space. However, some important aspects of the *in vivo* milieu like temperature, low-aerobic conditions, and host immunity were not represented in our model. Although occasionally we observed isolates with eccentric growth curve shapes, our impression is that the derivatives still well represented the different growth features. In contrast to an earlier report (31), in rich medium we also identified marked differences in lag phase

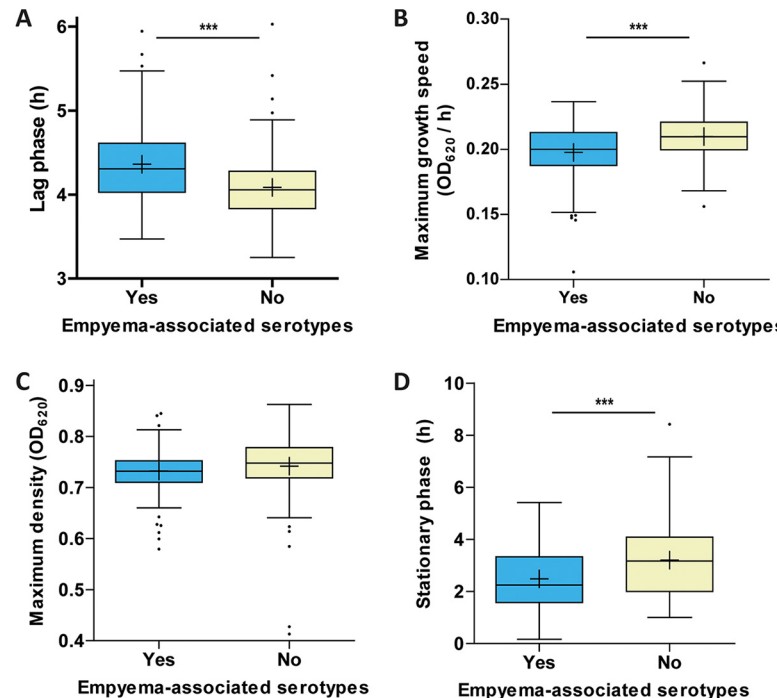

**FIG 6** Serotypes epidemiologically associated with empyema. Derived growth features for 379 *S. pneumoniae* blood culture isolates stratified according to the serotypes' epidemiological association with pleural empyema (yes, *n* = 204, blue; no, *n* = 175, yellow). Distributions are visualized in either Tukey box plots with mean (+) and outliers (·). The displayed growth features are lag phase (A), growth speed (B), maximum density (C), and stationary phase (D). $OD_{620}$, optical density at 620 nm; *, *P* < 0.017 (Bonferroni corrected significance threshold); **, *P* < 0.005; ***, *P* < 0.0005.

**TABLE 2** Tabular overview of differential growth features, as related to previously healthy adults infected[a]

| Clinical manifestation | Perspective | Serotypes involved | Lag phase | Growth speed | Maximum density | Stationary phase |
|---|---|---|---|---|---|---|
| Previously healthy adults | | | ↓ | ↑ | ↑ | ↓ |
| Mortality | Overall | | = | = | ↑ | = |
| Meningitis | Overall | | = | = | = | = |
| | Positively associated serotypes | [b] | = | ↑ | ↑ | ↓ |
| | Within serotype | 6A, 18C, 23F | ↓ | | | |
| | | 6B, 8, 18C, 23F | | | ↑ | |
| | | 1, 3, 23A | | | | ↓ |
| | Negatively associated serotypes | 4, 14(9) | = | = | ↓ | ↑ |
| Pleural empyema | Overall | | = | = | = | = |
| | Positively associated serotypes | 1, 3, 7F, 8, 9V, 14(9), 19A | ↑ | ↓ | = | = |
| | Within serotype | 7F, 14(9), 14(124), 19A, 23F | | | | ↑ |
| Capsular switch | In TIGR4 background | [c] | ± | ↓ | ± | ↓ |

[a]↓, growth feature value relatively low; =, growth feature value intermediate; ↑, growth feature value relatively high.
[b]Serotypes 6A, 6B, 6C, 7F, 10A, 15B, 19F, 23A, 23B, and 23F.
[c]Serotypes 1, 2, 3, 4, 5, 6A, 6B, 7F, 8, 9A, 9B, 9C, 11A, 12F, 18C, 19A, 19F, 23F, and 35B.

between pneumococcal isolates. A possible explanation is that we measured the time to relative change in the individual growth curve, instead of time to a fixed optical density (OD) for any strain in the earlier report. Furthermore, because culture history is known to be crucial for bacterial adaptation, we followed a highly standardized preculturing procedure for all strains tested (42).

The reproducibility of the measurements was very high. However, the duration of the stationary phase was relatively more variable, which was mainly due to variable initiation of the lytic growth phase—at the end of the stationary phase. If we compare our rankings of major disease serotypes to those reported for pneumococcal carriage isolates, similar patterns are observed for maximum density. However, differences are also present, suggesting that the validity of our growth measurements may be limited by growth conditions, data processing, or sampling population (37).

The relationship between high bacterial density and risk of mortality is in line with observations from human pneumococcal pneumonia (43). Although in more complicated pneumococcal infections, high bacterial loads in blood may result from continuous seeding from an unresolved site of infection (44), our current study suggests it may also be a predisposing factor for the development of meningitis but not empyema. How the optical density relates to the actual bacterial load remains uncertain. Some serotypes that were previously characterized as having a relatively thick capsule also demonstrated a relatively high maximum optical density in our *in vitro* assay, suggesting that a higher optical density does not necessarily represent a linearly increased number of cells. However, this relation was not universal and also within a serotype different lineages can display distinct maximum optical densities (45). In addition, in the host, within the different tissues, bacterial densities are affected by many different factors, with a large role for the immune system. It is difficult and beyond the scope of this report to translate our observed bacterial densities to actual densities within the host. Our aim was to examine the dynamics and kinetics of pneumococcal growth of the different isolates obtained from IPD patients and to relate this to the clinical phenotype.

Beyond serotype-based epidemiology, a limited number of studies have otherwise marked bacterial characteristics as determinants of the development of meningitis: *in vivo* studies comparing three serotypes and recent pneumococcal genome-wide association studies (21, 46–49). This study is among the first to systematically study pneumococcal growth phenotype as a covariate of human meningitis (50). Like we hypothesized, the isolates associated with meningitis showed relatively rapid and dense growth. Although it is still hard to interpret the relevance of the differences identified in our study, the observed pattern is in favor of our hypothesis that, in addition to possibly more sophisticated virulence factors, bacterial mass matters. A short lag phase was associated only with meningitis cases within certain serotypes. Furthermore, meningitis was associated with a relatively

short stationary phase up to the start of autolysis. Early during the stationary phase, pneumococci primarily produce lipoteichoic acids (LTA) in their cell walls, which halts autolysis (51). It is uncertain whether in our model there was a generic intrinsic trigger to switch toward wall teichoic acid (WTA), permitting autolysis. It is plausible, however, that early bacterial cell lysis in cerebrospinal fluid elicits a prominent inflammatory response and subsequent host damage.

Although empyema-associated serotypes showed a markedly longer lag phase, it is important to realize that this class was made up of more than half of all cases in the study. In contrast to meningitis, we observed no eccentric effects within a serotype to confirm the increased likelihood of establishing empyema due to a long lag phase or slow growth. The relatively long stationary phase, however, also observed for empyema cases within serotypes, is more likely to be causal for developing empyema. While pleural fluid is a potent growth medium, pneumococcal cells first need to cross mesothelial layers transcellularly and are later faced with starvation in an inflamed fibrotic environment (52, 53). Human plasminogen activation inhibitor 1 (PAI-1) mediates fibrosis and containment of pleural infection and is upregulated by LTA. This is why LTA is used for pleurodesis in malignant pleural effusion (54–56). Taken together, these facts indicate it is very possible that pneumococci that tend toward a long catabolic stationary phase, characterized by expression of LTA, can both induce and persist in the fibrotic pleural environment.

A capsule switch primarily affected the stationary phase, while lag phase and maximum density seemed to depend on additional factors. Vice versa, the two genetically distinct pneumococcal lineages (MLST 9 and 124) that both carry the serotype 14 capsule showed a similar stationary phase but dissimilar lag phases and maximum densities. An explanation for the stationary phase being heavily dependent on capsule type is that expression of the capsular locus is maximal early in the stationary phase (57) but no longer reflects capsule thickness (19). In the stationary phase, pneumococci catabolize their own capsular polysaccharide using the genes involved in its biosynthesis, facilitating a prolonged survival in stressful conditions (30). A switch of capsule affected the maximum growth rates, although all mutants seemed able to fully exhaust available nutrients to a similar maximum density at some point. These experimental observations are in agreement with the epidemiological notion that capsule type and genetic lineage are interdependent for efficient circulation (58).

If serotype-specific growth features remain static in the population, vaccine-induced serotype replacement could potentially drive changes in prevalence of certain disease manifestations. Non-PCV13 serotypes binned as one group displayed a relatively low maximum optical density and a relatively short stationary phase, features that do not fully match any of the disease manifestations studied here. However, the crude patterns observed in our current study likely originate from more pronounced serotype- or lineage-specific effects of particular growth features on the likelihood of dissemination in human disease. If growth features could accurately be modeled from pneumococcal genomics, this would provide a means to easily monitor and expand our knowledge on clinical risks associated with replacing serotypes.

This study provides a first exploration of differential pneumococcal growth properties in clinical disease. It makes us consider the possibility that whether bacterial properties mediate virulence is specific to the invasive site of infection. The reason that meningitis requires a relatively shorter course of antibiotics compared to pleural empyema may extend beyond host clearance. The bacterium's behavior at that site of infection may also play a role in treatment response (59).

Important to realize is that the shape of a growth curve could well be determined by only a fraction of the bacterial population. For example, the end of the lag phase can be determined by a single cell that starts exponential growth. In what state is the remainder of the original infecting population, and how important is that for the shape of the growth curve, for the development of complicated infections, and for the treatment required?

Although these questions require further investigation, the growth curves are probably representative for the potential of the infecting population.

The model system that we used for studying culture dynamics does not represent bacterial kinetics in relation to the host immune system, while it is this interaction that generally determines the course of an infection (60). Therefore, it is important to understand the mechanisms behind the robust growth features that we observed. Such knowledge can be used to manipulate bacterial growth phenotypes and to study their proportionality in more representative experimental models like *ex vivo* human specimens or an *in vivo* infection model. If that would provide support to the hypothesis that vigorous or indolent growth mediates different disease manifestations, this could ultimately lead to differential prevention and treatment strategies for the different metastatic infection sites that can complicate IPD.

## MATERIALS AND METHODS

**Pneumococcal isolates and clinical data.** A total of 383 clinical strains of *S. pneumoniae* comprising 33 serotypes were retrieved from the Pneumococcal Bacteraemia Collection Nijmegen (PBCN) cohort (61). They were isolated from patients diagnosed with pneumococcal bacteremia admitted to two Dutch hospitals between January 2000 and June 2011.

Serotyping was initially performed using multiplex PCR analysis according to Pai et al. (62). In case multiplex PCR was inconclusive, serotyping was performed by Quellung using Pneumococcus Neufeld Antisera (Statens Serum Institut, Copenhagen, Denmark) according to the manufacturer's instructions. The serotype was confirmed by genetic prediction from the capsular locus sequence using SeroBA (63). Antibiotic resistance was uncommon in our cohort: 1.3% penicillin, 1.8% doxycycline, and 0.8% macrolide resistance, respectively.

Details on the collection of the clinical variables were previously described (21). Briefly, the clinical manifestation of pneumonia was based on the attending physician's discretion, while the assignment of meningitis and pleural empyema were invariably supported by microbiological and/or biochemical test results of specimens collected from the site of infection. The clinical syndromes were not mutually exclusive. Solitary pneumonia was defined as no secondary foci of infection. Uncomplicated pneumonia was defined as solitary pneumonia without admission to an intensive care unit (ICU) or in-hospital 30-day mortality. Clinical phenotype and 30-day mortality were known for 96 and 95% of the cases, respectively. The Charlson comorbidity index score could be established for 362 out of 378 cases (96%). Cases with missing data were omitted solely from analyses that involved the missing variable concerned. The cohort chiefly concerns adults, for whom there was no national pneumococcal immunization program in place, except for very narrow indication.

**Pneumococcal mutants.** The capsular operon deletion mutants ($\Delta cps$) were constructed in *S. pneumoniae* laboratory strains TIGR4 (serotype 4), D39 (2), G54 (19F), PBCN0162 (18C), and PBCN0229 (5) as previously described by Pearce et al. (64). Isogenic capsular variants of *S. pneumoniae* strain TIGR4 were constructed and kindly provided by Trzciński et al. (65).

**Pneumococcal growth curve.** Frozen bacterial inoculum was prepared by overnight culture on Columbia 5% blood agar, from which a few colonies were inoculated in liquid medium (50% M17, 50% casamino acids tryptone [CAT] medium, 0.25% glucose) for growth, all at 37°C and 5% $CO_2$. At $OD_{620}$ 0.29 to 0.31 glycerol was added (end concentration, 16%), and aliquots were stored at $-80$°C. Growth kinetics at 37°C and 5% $CO_2$ were measured every 10 min over 15 h at $OD_{620}$ by microplate reader (Spark 10M, Tecan, Switzerland) with use of a humidity cassette. In each well of a sterile flat-bottomed 48-well plate (Nunclon Surface, Nunc, Denmark), 15 $\mu$L of inoculum was added to 1.5 mL of prewarmed rich growth medium (45% M17, 45% CAT, 0.225% glucose, 10% fetal calf serum [Greiner Bio-one], 26 U/mL catalase [Sigma-Aldrich C1345]). The growth medium was supplemented with catalase, which is protective of $H_2O_2$-mediated killing of *S. pneumoniae* and which is available in blood (66). For each isolate, six repeat measurements were performed on three separate days.

**Derived growth features.** From the raw measurement data, the metrics of the growth feature were calculated using a customized program in R. The end of the lag phase was defined as the first time point at which the baseline $OD_{620}$ (which equals the average of measurement 6 to 10) had increased $\geq 1.5$-fold. The maximum growth speed was calculated from the slope of the linear part of the $\log_2$ transformed growth data and expressed as increments in $OD_{620}$/h. The maximum density was calculated from the mean of the highest $OD_{620}$ values. The start of the stationary phase was defined as the time point at which the maximum OD was reached, and the end of the stationary phase was defined as the time point at which the OD dropped below 95% of the maximum OD.

**Data analysis.** For further analyses of growth features, the average of 6 repeat measurements was taken for lag time, maximum growth speed, and maximum density. As repeat measurements of stationary phase were less normally distributed, their median was used. Principal component analysis of the four derived growth features was performed via online tool Clustvis (67). Relatedness between pairs of growth features were assessed by linear regression. "Pathogenic poles" of growth features were identified by probing which end of the distribution affected relatively healthy patients. The extremity of the growth feature at which healthy individuals were also affected was deemed as the pole with the highest pathogenic potential. Corresponding differences in growth features were tested one-sided by *t* test or

ANOVA, except for differences in stationary phase, which were tested by Mann-Whitney U or Kruskall Wallis tests. Serotype 14 was split according to MLST, because of its genotypically distinct lineages. Furthermore, serotypes were categorized according to their epidemiological propensity for causing clinical phenotypes as stated in the introduction (according to references 20–28). Multivariate logistic regression analysis was performed for modeling lethal infection, using backward likelihood-excluding variables without significant contribution to the model. The threshold for statistical significance was set at 0.05. Bonferroni correction for the growth features of serotypes that were tested for their association with the three clinical phenotypes resulted in a threshold of 0.017.

**Data availability.** Data and visualization of individual growth curves are publicly accessible via the online interactive database: https://fairdb.tenwiseservice.nl/GrowthViewer/.

## SUPPLEMENTAL MATERIAL

Supplemental material is available online only.

**SUPPLEMENTAL FILE 1**, PDF file, 1.2 MB.

## ACKNOWLEDGMENTS

We acknowledge the participating hospitals and affiliated researchers for their contributions and support. We kindly thank K. Trzciński for providing *S. pneumoniae* capsular switch mutants.

This study was supported by TARGET project grant JPIAMR2019-087 funded by JPI-AMR–ZonMW, The Netherlands.

This observational cohort study was approved by the local medical ethics committees of both participating hospitals.

We declare no conflict of interest.

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
