## [Reviewer comments · Microbiology Spectrum]

Microbiology Spectrum

Differential pneumococcal growth features in severe invasive disease manifestations

Daan Arends, Wynand Alkema, Indri Putri, Christa van der Gaast - de Jongh, Marc Eleveld, Jeroen Langereis, Quirijn de Mast, Jacques Meis, Marien de Jonge, and Amelieke Cremers

Corresponding Author(s): Amelieke Cremers, Radboudumc Center for Infectious Diseases

Review Timeline:

Submission Date:	January 5, 2022
Editorial Decision:	March 8, 2022
Revision Received:	May 9, 2022
Accepted:	May 23, 2022

Editor: Joanna Goldberg

Reviewer(s): Disclosure of reviewer identity is with reference to reviewer comments included in decision letter(s). The following individuals involved in review of your submission have agreed to reveal their identity: Tulip A Jhaveri (Reviewer #2)

Transaction Report:

DOI: <https://doi.org/10.1128/spectrum.00050-22>

March 8, 2022

Dr. Amelieke Johanna Hendrina Cremers
Radboudumc Center for Infectious Diseases
Department of Medical Microbiology
PO Box 9101
Internal post 777
Nijmegen 6500 HB
Netherlands

Re: Spectrum00050-22 (Differential pneumococcal growth features in severe invasive disease manifestations)

Dear Dr. Amelieke Johanna Hendrina Cremers:

Thank you for submitting your manuscript to Microbiology Spectrum. You will see that both reviewers have many comments.

I also carefully read your paper and have additional suggestions that I believe will help with the readability of the manuscript:
--I think it would be helpful to present references describing the correlation of the serotypes with the various types of infections in the Introduction. Then if this does hold up in the Results observed here, it would be more readily apparent.
--Also, the importance and utility of the various "experimental mutants (line 140)" is never clearly presented. Are these the same as "artificial capsule variants (line 133)"?
--I do not understand the term "Pathogenic Pole". This should be defined.
--Also, I believe one of the reviewers mentioned this, but I would like to echo, the AMOUNT of capsule was not determined. Could that be an added variable? Do certain serotypes naturally produce more or less capsular polysaccharide?

Link Not Available

Sincerely,

Joanna Goldberg

Journals Department
Reviewer comments:

Reviewer #1 (Comments for the Author):

In this study, Arends et al investigated the growth features for 378 *S. pneumoniae* blood isolates and 25 experimental mutants. The growth features were then related to the propensity of causing severe manifestations of invasive pneumococcal disease. The topic is relevant, interesting and the chosen methods are technically sound. However, I think the authors should make clearer the limitations of the study as there are some concerns (see below):

Limitations/concerns

Antibiotic resistance: Did the authors check for associations of the growth features with the antibiogram of the strains? It has been hypothesized that there is a fitness cost for antibiotic resistant strains. If true, this would be relevant in this study. Could this perhaps be added to the study?

Lack of sequencing data: It is a limitation that no whole genome sequencing or MLST data are shown. As the authors correctly point out, both, geno- and serotype may matter. The inclusion of mutants in the TIGR4 background is a good idea, but, unfortunately, these are not clinical isolates. As far as I understand from PMID: 29788414, sequencing data would be available? The growth conditions look very fine to me. However, I would recommend to also test some different growth conditions. It would be interesting to know if the growth differences are maintained in different media mimicking the blood (i.e. rich media). Not including different conditions could be added as a limitation in the discussion section.

Different serotypes have different capsule thickness. Though not to a large extent, this may affect the OD measurements at times and CFU counting would be more accurate. I suggest discussing the relevance of the varying capsule thickness of the OD measurements in the discussion section.

Some clinical data is missing. What about the vaccination status? This would be a very important variable to consider.

Vaccinated individuals are more prone to be infected with a non-PCV serotype. Non-PCV serotypes might be less virulent and maybe this is also shown as a varying growth feature?

Minor comments:

Line 317: The authors are sure concerning citing '14'? Double check citations.

Line 365: Serotype 14 was split according to MLST. What about the MLST of all the other strains?

Line 367: I appreciate that the authors performed Multivariate logistic regression analysis (e.g. table 1). However, it is unclear at times how this has been used. E.g. have the statistical analyses in figure 3 and figure 6 been corrected with the clinical metadata like age etc.?

Reviewer #2 (Comments for the Author):

Thank you for this opportunity to review this study. Arends and colleagues have evaluated growth features of 378 blood pneumococcal isolates (including 33 serotypes) and 25 experimental mutants to gain insight into the pathogenesis of invasive pneumococcal disease (IPD). The manuscript summarizes differential growth characteristics, particularly the lag phase, growth speed, maximum bacterial density and stationary phase, and its relation to specific clinical manifestations, mortality, and serotypes of IPD.

Overall, the manuscript is clear and sound, methods are well described, data is robust and the analysis produces some rich results backed up by clear tables, figures and discussion points. Previous studies have described the role of pneumococcal serotypes (PMID: 23341706) and host factors (PMID: 23341706) in the pathogenesis of IPD. However, very few studies, such as this, systematically describes the role of major pneumococcal growth kinetics in predicting IPD. Authors, in the discussion, have acknowledged the limited scope of the study as far as some unanswered questions go, yet have been able to tie the in vivo and in vitro data as closely as possible. While this paper has certainly been engaging, some issues in this paper require explanation. Please see comments below for specific concerns.

Major comments

-Were all the clinical manifestations (empyema/meningitis, etc) described in the analysis per attending physician's discretion or were positive culture data from non-blood specimens taken into consideration? Since all the isolates obtained were blood cultures, authors may consider enlisting how many of these had concomitant positive CSF/respiratory cultures for *S. pneumoniae*. Were there any cases that had more than one clinical manifestation (for eg. Both meningitis as well as empyema)?
-It would be helpful to clarify how serotypes were detected under methods section.

Minor comments

Line 32: remove the word 'body'

Line 53: I believe *M. tuberculosis* is the leading pathogen causing deaths worldwide, and pneumonia (Lower resp tract infection) is the leading infectious cause of deaths worldwide. *Strep pneumoniae* is incorrect; if the authors believe it is true, it should be supported with a reference

Line 176: 'TIGR4' is abruptly included - please provide a background what this entails.

Lines 316-322: these can go to Results section.

Line 320: for 96% and 95% 'of the cases' respectively.

Line 323: please explain this sentence - 'details on data collection of the cohorts were described elsewhere'

Table 1- please clarify if it was adjusted or unadjusted OR

Table 2- spell 'previously' completely instead of 'prev'

Staff Comments:

Preparing Revision Guidelines

Please return the manuscript within 60 days; if you cannot complete the modification within this time period, please contact me. If you do not wish to modify the manuscript and prefer to submit it to another journal, please notify me of your decision immediately so that the manuscript may be formally withdrawn from consideration by Microbiology Spectrum.

Spectrum00050-22 Response to Reviewers

In our response we refer to the lines in the Marked-up version of the manuscript.

Editor Goldberg

--I think it would be helpful to present references describing the correlation of the serotypes with the various types of infections in the Introduction. Then if this does hold up in the Results observed here, it would be more readily apparent.

We agree with the editor that it should be more clearly stated that references 20-28 (in the Introduction) support the population-based correlations between serotype and type of infection. We have therefore added this explicitly to the methods section. References 45-47 in the discussion section comprise the few related and otherwise oriented studies available (anecdotal *in vivo* meningitis model, and study on invasiveness [not type of infection] in a meningitis cohort). We have now adjusted the sentence to clarify its function (lines 263-267).

--Also, the importance and utility of the various "experimental mutants (line 140)" is never clearly presented. Are these the same as "artificial capsule variants (line 133)"?

Indeed experimental mutants refers to the artificial capsule switch mutants as well as the capsule deletion mutants, as described in the 'serotype-based effects' paragraph in the results section lines 170-179 and referred to in line 131-132. We have added this clarification in line 139.

--I do not understand the term "Pathogenic Pole". This should be defined.

With 'pathogenic pole' we mean the side of the distribution of the bacteria within the growth feature data at which the bacteria have the highest potential for causing disease. At these extremities, previously healthy adults also get IPD, whereas at the opposite site of the spectrum predisposed individuals suffer IPD. We have added a line (414-416) further explaining which pole we define as more pathogenic.

--Also, I believe one of the reviewers mentioned this, but I would like to echo, the AMOUNT of capsule was not determined. Could that be an added variable? Do certain serotypes naturally produce more or less capsular polysaccharide?

There is evidence for certain serotypes to produce more or less capsule (PMID: 19521509). Below is a figure showing maximum optical density measured for different serotypes with differently sized capsule thicknesses, ordered from thickest to thinnest capsule according to the above reference. It illustrates that thickness of capsule does not linearly affect the maximum optical density. It also shows that intraserotype differences are observed, as serotype 14 shows a different OD-max, depending on its genetic background. We added a line (255-259) to the discussion regarding this observation. However, we need to be cautious with conclusions on the amounts and thickness of capsules. The gold standard methods is electron microscopy which is a very low throughput measure while it is know that there is biological variation that has to be taken into account and for which a high throughput method is needed. The high throughput measures used (ELISA or FITC-dextran exclusion) are very hard to reproduce as we recently experienced in our laboratory. Furthermore, it is

not unlikely that amounts and thickness of capsule are influenced by growth conditions. However, we can only confirm this when the appropriate methods are refined.

Reviewer comments:

Reviewer #1 (Comments for the Author):

In this study, Arends et al investigated the growth features for 378 *S. pneumoniae* blood isolates and 25 experimental mutants. The growth features were then related to the propensity of causing severe manifestations of invasive pneumococcal disease. The topic is relevant, interesting and the chosen methods are technically sound. However, I think the authors should make clearer the limitations of the study as there are some concerns (see below):

Limitations/concerns

Antibiotic resistance: Did the authors check for associations of the growth features with the antibiogram of the strains? It has been hypothesized that there is a fitness cost for antibiotic resistant strains. If true, this would be relevant in this study. Could this perhaps be added to the study?

We thank the reviewer for this comment. Mainly beta-lactam resistance has been associated with fitness cost in *S. pneumoniae*. However, in this cohort from the Netherlands only 5 isolates were not fully susceptible to penicillin. Doxycycline resistance (n=7 isolates) and macrolide resistance (n=3 isolates) were also rare. This is now mentioned in the methods section (line 362-363).

Lack of sequencing data: It is a limitation that no whole genome sequencing or MLST data are shown. As the authors correctly point out, both, geno- and serotype may matter. The inclusion of mutants in the TIGR4 background is a good idea, but, unfortunately, these are not clinical isolates. As far as I understand from PMID: 29788414, sequencing data would be available?

Indeed draft genomes are available for most of the strains in the dataset. We have now added the known MLST types to the online growth viewer, where we will refer to the publicly available bacterial genome data. We agree that it is interesting to study genetic determinants of pneumococcal growth features in wild type pneumococci. These analyses are currently being performed together with an experienced collaborator, and will include validation experiments in a second set of 600 IPD isolates that we collected and sequenced by now. We think this is, however, beyond the scope of the current manuscript that focuses on the clinical relevance of bacterial growth features.

The growth conditions look very fine to me. However, I would recommend to also test some different growth conditions. It would be interesting to know if the growth differences are maintained in different media mimicking the blood (i.e. rich media). Not including different conditions could be added as a limitation in the discussion section.

The medium we used for determining growth differences was specifically selected to mimic the spectrum of invasive infection using very rich and serum-supplied medium mimicking bloodstream infection as explained in lines 385-395,231-242. The most interesting model for validation of our results would be human specimens from relevant infection sites. However, such complex media that allow variable penetration of light, are not suited for conducting high-throughput optical density measurements of bacterial growth that were currently applied.

Different serotypes have different capsule thickness. Though not to a large extent, this may affect the OD measurements at times and CFU counting would be more accurate. I suggest discussing the relevance of the varying capsule thickness on the OD measurements in the discussion section.

Kindly also see our response to the editor's remarks of similar nature. A line was added to the discussion section (255-259).

In addition, the experience with CFU counts is that these often do not compare well between serotypes with different colony morphologies, in terms of the number of original cells it represents.

Alternative measures for viable cell counts could be microscopy or FACS, however we feel this is beyond the scope of the current project.

Some clinical data is missing. What about the vaccination status? This would be a very important variable to consider. Vaccinated individuals are more prone to be infected with a non-PCV serotype. 1. Handling of missing data is now described in the methods section (lines 372-373). The clinical scores are not validated for use in children.

2. Vaccination status. The reviewer wonders whether a PCV13-immunised status could be a confounder of the relation between adults at risk and “less virulent” growth features. Hereby suggesting that a) non-PCV serotypes are “less virulent” in general and b) this group would (if vaccinated) be more likely to be infected by non-PCV serotypes.

a) In children serotype replacement is modest, indeed suggesting that non-PCV serotypes may be less likely to cause invasive disease. However, for adults in the Netherlands there was complete serotype replacement of IPD after introduction of PCV7 in the pediatric immunisation programme (in 2006, cohort 2000-2011), and we observed that the severity of disease caused by vaccine vs. non-vaccine serotypes altered over time (PMID: 24814555) and may therefore not be static. Still, we compared the observed growth features between PCV-13 and non-PCV-13 serotypes and elaborated on potential effects on clinical manifestations in the future (supplementary figure 8, lines 209-212 and 305-313). Non-PCV-13 serotypes did not universally display more indolent growth.

b) During the study period there was not yet a pneumococcal vaccination programme in place for adults in the Netherlands, except from a few narrow indications (primarily asplenia). Hardly any patient in this cohort had been vaccinated. Non-PCV-13 serotypes were indeed equally present in healthy adults (24%) and adults at risk (28%).

Based on these observations we believe that an immunised status has not been a relevant confounder in the current cohort. The reviewer’s remarks motivated us however to provide more information on the background and composition of the cohort (lines 351-376 and 154-157).

Minor comments:

Line 317: The authors are sure concerning citing '14'? Double check citations.

We thank the reviewer for pointing out this error. The cohort is described in PMID: 24814555, and the data collection in PMID: 29788414. All other citations throughout the manuscript are correct.

Line 365: Serotype 14 was split according to MLST. What about the MLST of all the other strains?

We will add the MLST data to the online growth-viewer, as mentioned previously.

Line 367: I appreciate that the authors performed Multivariate logistic regression analysis (e.g. table 1). However, it is unclear at times how this has been used. E.g. have the statistical analyses in figure 3 and figure 6 been corrected with the clinical metadata like age etc.?

The multivariate logistic regression was only applied once: modelling contributors to mortality. This information is added to the methods section (line 422).

Reviewer #2 (Comments for the Author):

Thank you for this opportunity to review this study. Arends and colleagues have evaluated growth features of 378 blood pneumococcal isolates (including 33 serotypes) and 25 experimental mutants to gain insight into the pathogenesis of invasive pneumococcal disease (IPD). The manuscript summarizes differential growth characteristics, particularly the lag phase, growth speed, maximum bacterial density and stationary phase, and its relation to specific clinical manifestations, mortality, and serotypes of IPD.

Overall, the manuscript is clear and sound, methods are well described, data is robust and the analysis produces some rich results backed up by clear tables, figures and discussion points. Previous studies have described the role of pneumococcal serotypes (PMID: 23341706) and host

factors (PMID: 23341706) in the pathogenesis of IPD. However, very few studies, such as this, systematically describes the role of major pneumococcal growth kinetics in predicting IPD. Authors, in the discussion, have acknowledged the limited scope of the study as far as some unanswered questions go, yet have been able to tie the in vivo and in vitro data as closely as possible. While this paper has certainly been engaging, some issues in this paper require explanation. Please see comments below for specific concerns.

Major comments

-Were all the clinical manifestations (empyema/meningitis, etc) described in the analysis per attending physician's discretion or were positive culture data from non-blood specimens taken into consideration? Since all the isolates obtained were blood cultures, authors may consider enlisting how many of these had concomitant positive CSF/respiratory cultures for S. pneumoniae. Were there any cases that had more than one clinical manifestation (for eg. Both meningitis as well as empyema)?

The clinical manifestation of pneumonia was based on the attending physician's discretion, while the assignment of meningitis and pleural empyema were invariably supported by local biochemical and/or microbiological test results (line 364-368 methods section). While clinical syndromes were not mutually exclusive, none of the patients had pleural empyema in combination with meningitis (line 156-157).

-It would be helpful to clarify how serotypes were detected under methods section.

Serotyping was initially performed using multiplex PCR analysis according to Pai et al. (PMID: 16390959). In case multiplex PCR was inconclusive, serotyping was performed by Quellung using Pneumococcus Neufeld Antisera (Statens Serum Institut, Copenhagen, Denmark) according to the manufacturer's instructions. The serotype was confirmed by genetic prediction from the capsular locus sequence using SeroBA (PMID: 29870330). This was added to the methods section line 358-361.

Minor comments

Line 32: remove the word 'body'

We have removed 'body'.

Line 53: I believe M. tuberculosis is the leading pathogen causing deaths worldwide, and pneumonia (Lower resp tract infection) is the leading infectious cause of deaths worldwide. Strep pneumoniae is incorrect; if the authors believe it is true, it should be supported with a reference

Annual deaths from pneumococcal pneumonia alone are estimated at >1.5 million (PMID: 28843578), which is higher than TB, and therefore we believe it is fair to call it a leading cause. If we are not mistaken, there are however no references allowed in the "Importance" section.

Line 176: 'TIGR4' is abruptly included - please provide a background what this entails.

We have altered line 176 to provide background for TIGR4.

Lines 316-322: these can go to Results section.

We have moved the content of lines 319-321 to the results section, but have chosen to keep the other data in the methods section available for readers with particular interest.

Line 320: for 96% and 95% 'of the cases' respectively.

We have edited the sentence, which can now be found in line 370-371

Line 323: please explain this sentence - 'details on data collection of the cohorts were described elsewhere' We have now written out the most relevant details in the methods section (lines 364-376).

Table 1- please clarify if it was adjusted or unadjusted OR

Table 1 shows the adjusted ORs for death from IPD, after multivariate logistic regression. For clarification, it was adjusted in the title and the header of Table 1.

Table 2- spell 'previously' completely instead of 'prev'

We have edited it accordingly

May 23, 2022

Dr. Amelieke Johanna Hendrina Cremers
Radboudumc Center for Infectious Diseases
Department of Medical Microbiology
PO Box 9101
Internal post 777
Nijmegen 6500 HB
Netherlands

Re: Spectrum00050-22R1 (Differential pneumococcal growth features in severe invasive disease manifestations)

Dear Dr. Amelieke Johanna Hendrina Cremers:

Your manuscript has been accepted, and I am forwarding it to the ASM Journals Department for publication. You will be notified when your proofs are ready to be viewed.

Sincerely,

Joanna Goldberg
Editor, Microbiology Spectrum

Journals Department
1: Accept